# Differences in Free-Living Patterns of Sedentary Behaviour between Office Employees with Diabetes and Office Employees without Diabetes: A Principal Component Analysis for Clinical Practice

**DOI:** 10.3390/ijerph191912245

**Published:** 2022-09-27

**Authors:** Francesc Alòs Colomer, Mª Àngels Colomer Cugat, Judit Bort-Roig, Emilia Chirveches-Pérez, Yoseba Cánovas Zaldúa, Carlos Martín-Cantera, Josep Franch-Nadal, Anna Puig-Ribera

**Affiliations:** 1Primary Healthcare Centre Passeig de Sant Joan, Catalan Health Institute, 08010 Barcelona, Spain; 2Member of the redGDPS Foundation, 08204 Sabadell, Spain; 3Department of Mathematics, ETSEA, University of Lleida, 25198 Lleida, Spain; 4Sport and Physical Activity Research Group, Centre for Health and Social Care Research, University of Vic-Central University of Catalonia, 08500 Vic, Spain; 5Research Group on Methodology, Methods, Models and Outcomes of Health and Social Sciences, Centre for Health and Social Care Research, University of Vic-Central University of Catalonia, 08500 Vic, Spain; 6Head of Training, Catalan Health Institute, 08006 Barcelona, Spain; 7Barcelona Research Support Unit, Foundation Primary Care Research Institute IDIAP Jordi Gol, 08025 Barcelona, Spain; 8CIBER of Diabetes and Associated Metabolic Disease (CIBERDEM), Instituto de Salud Carlos III (ISCIII), 28029 Madrid, Spain

**Keywords:** sedentary behaviour, type 2 diabetes, principal component analysis, desk-BASED job, activPAL, disease management, sitting time

## Abstract

Aims: To identify principal components of free-living patterns of sedentary behaviour in office employees with type 2 diabetes (T2D) compared to normal glucose metabolism (NGM) office employees, using principal component analysis (PCA). Methods: 213 office employees (*n* = 81 with T2D; *n* = 132 with NGM) wore an activPAL inclinometer 24 h a day for 7 consecutive days. Comparions of sedentary behaviour patterns between adults with T2D and NGM determined the dimensions that best characterise the sedentary behaviour patterns of office employees with T2D at work, outside work and at weekends. Results: The multivariate PCA technique identified two components that explained 60% of the variability present in the data of sedentary behaviour patterns in the population with diabetes. This was characterised by a fewer number of daily breaks and breaks in time intervals of less than 20 min both at work, outside work and at weekends. On average, adults with T2D took fewer 31 breaks/day than adults without diabetes. Conclusion: Effective interventions from clinical practice to tackle prolonged sedentary behaviour in office employees with T2D should focus on increasing the number of daily sedentary breaks.

## 1. Background

T2D is one of the public health problems of the 21st century, affecting 537 million adults aged between 20 and 79 years (10.5%) in the world population, causing several million deaths every year and generating a high social and healthcare cost [1]. According to the International Diabetes Federation (IDF, 2021), a substantial increase is expected for the coming years, with an estimated 783 million (12.2%) people being affected by 2045 [1]. The high prevalence of T2D is multifactorial, and the main driving factors include overweight, obesity and changes in lifestyle related to sedentary behaviour (SB) [2].

SB is defined as any waking behaviour while in a sitting, reclining or lying posture characterised by an energy expenditure ≤ 1.5 times the basal metabolic rate or METs (Metabolic Equivalent Task) [3]. The SB pattern is defined as the manner in which SB is accumulated throughout the day while awake, including the total sedentary time, the number of sedentary interruptions/breaks daily (a non-sedentary bout in between two sedentary bouts), the frequency and duration of the sedentary bouts (a period of uninterrupted sedentary time), and the time accumulated in each period [3]. SB has a high prevalence in adults with abnormal glucose metabolism and T2D [4] and is considered an important modifiable factor for the prevention, development and first-line therapeutic management of patients with T2D [5,6].

In people with T2D, long periods of time spent in SB are associated with a poorer metabolic profile [7]. Breaking up prolonged sedentary time with brief, regular and frequent bouts of light-intensity physical activity is associated with an improvement in cardio metabolic profile, a reduction in blood pressure, and attenuation of acute postprandial glucose, insulin, C-peptide and triglyceride responses [7]. Accordingly, current physical activity and SB guidelines recommend substituting as much sedentary time as possible for light-intensity physical activity and interrupting sedentary time with breaks of light-intensity walking or simple resistance activities [6]. However, they do not provide specific recommendations for the population with diabetes or for those with general chronic pathology regarding the duration and frequency with which these sedentary periods should be interrupted [6]. In this context, addressing in clinical practice the SB pattern in people with diabetes is not widespread, traditionally focusing on increasing moderate and vigorous physical activity [8].

Characterising the SB pattern in patients with T2D would provide us with information to individualise the approach to SB in the population with diabetes and develop effective interventions from clinical practice that mitigate the health risks associated with prolonged sedentary time. To our knowledge, there are scant data on how SB is patterned among adults with T2D, especially regarding the length and accumulation of sedentary bouts using the gold standard measure for SB (activPAL device) [7,9]. The activPAL device is a thigh-worn sensor that provides inclinometer-derived information to determine time spent on body postures (i.e., sitting/lying and upright) and transitions between these postures. It shows excellent agreement with direct observation for sitting/lying time, upright time, sitting/lying to upright transitions and for detecting reductions in sitting [10]. 

Such data is needed to inform specific recommendations and interventions on how to reduce and break up prolonged SB in the prevention and management of T2D. Given that patients with T2D report the most frequent visits with primary care providers [11], characterising these patients’ free-living patterns of SB is a key issue for effectively tackling SB from primary care settings. On the need to provide a comprehensive understanding of free-living patterns of SB in clinical populations, PCA analysis can simplify sets of data with a high number of variables to identify what best characterizes that set of data.

The aim of this study was to use PCA to examine activPAL-based free-living patterns of SB in office employees with T2D compared with office employees without diabetes. This formative research will contribute towards translational research by (i) facilitating the identification of differences and similitudes between adults with and without diabetes to best characterise the free-living patterns of SB in adults with T2D; (ii) helping primary care providers focus on SB patterns tailored to this clinical population; and (iii) facilitating the identification of potential targets for behavioural interventions on SB for people with T2D, including frequency of interruptions and length of sedentary bouts.

## 2. Methods

### 2.1. Study Design

A cross-sectional study. The inclusion criterion for the group of T2D patients was to be diagnosed with T2D in accordance with international criteria [12]. The rest of the inclusion criteria were common for both groups, participants with diabetis and without diabetes: patients between 18 and 65 years old (working age); office employees with a minimum of 55% of their daily working hours performing sedentary tasks according to the Occupational Sitting and Physical Activity Questionnaire (OSPAQ) [13]; and having a work contract of at least 18.5 h/week. 

The exclusion criteria for both groups consisted of (i) having a diagnosis of musculoskeletal, cardiovascular, pulmonary or orthopedic problems or any other physical condition that prevented them from being physically active; (ii) participating simultaneously in another study or programme of sedentary behaviour, physical activity, nutrition or weight control; (iii) being pregnant; or (iv) having a history of psychiatric problems or substance abuse that could interfere with adherence to the study protocol.

### 2.2. Setting

The group of participants with normal glucose metabolism (NGM) were a convenience sample of administrative staff (*n* = 132) recruited from four Spanish hospitals to take part in an mhealth intervention to “sit less and move more” during working hours [14]. The study was approved by the ethics committee of each hospital [14] and the recruitment procedure is detailed elsewhere [14]. 

The group of participants with T2D (*n* = 81) were recruited at primary care centres in Barcelona between April 2019 and January 2020 with the aim of receiving a medical prescription offered by an mhealth programme to reduce and interrupt sedentary time at work. In each centre and during the recruitment process, the professionals involved (general practitioners and nurses) identified and recruited patients in medical consultations through the health centre’s database after verifying that they met the inclusion criteria. Written informed consent was obtained from all participantsincluded in the study. This trial was approved by the IDIAP Jordi Gol Clinical Research Ethics Committee with registration number P18/102. The recruitment procedure is detailed elsewhere [15].

### 2.3. Variables and Data Measurement

The activPAL3TM device (PAL Technologies Ltd., Glasgow, UK) measured and quantified the physical activity and SB of patients with T2D and NGM desk-based employees. This device has already been demonstrated as a valid measure to quantify body posture and activity patterns during free-living conditions [16]. The device was attached to the participants’ right thigh using a flexible nitrile sleeve and a transparent film (10 × 10 cm of hypoallergenic Tegaderm™ Foam Adhesive Dressing). The waterproof dressing of the activPAL3TM allowed participants to wear the monitor continuously for 24 h per day for 7 complete days. Participants received additional dressings and instructions on how to reattach the device if needed. Additionally, participants were asked to record their daily wake-up time, bedtime, working hours, and any monitor removal time.

Data were processed using activPAL Professional Software™ (version 7.2.32), Microsoft Excel 2010 (Redmond, WA, USA), and MATLAB v8.4 (MathWorks^®^, Natick, MA, USA), following previously published protocols [16]. From the activPAL3TM software output, the following outcomes were determined: total sitting time, total standing time, total number of sitting bouts and number of sitting bouts with four different lengths (<20 min, 20–40 min, 40–60 min and >60 min). Additionally, total time spent in light-intensity physical activity (1.5–3 METs) and moderate-to-vigorous physical activity (defined as at least >3 METs) was determined by using previously validated count-to-activity thresholds [16]. Overall, the outcomes were reported as averages of the total week, weekdays (working and non-working hours), and weekend days. Wake up time, bedtime, working and non-working hours and device removal time were recorded by using the participants’ paper diary.

Furthermore, age, sex and anthropometric variables by weight, height and body mass index (BMI) were recorded for each participants. The anthropometric variables were measured with a Seca 770 scale and a Seca 222 height measuring rod. The BMI was calculated by dividing the body weight by the square of the height in metres (kg/m^2^). The set of variables measured for the study was 57, 53 related to the SB pattern and physical activity and the other ones are: participants with diabetes, age, sex and BMI (Appendix A).

### 2.4. Sample Size

The sample size was initially set at 132 patients with T2D and 132 adults with NGM. We contacted 160 patients with T2D that met the inclusion criteria, considering a 20% loss. Forty-two who did not respond, 20 that were not interested and 17 that failed to record correctly the measures taken by the activPAL device (insufficient record of days or errors in the record) were excluded. The total number of patients with T2D with the record of minimum days set was 81. The sample total was 213 participants. Using the free software G*Power [17] for a comparison test of bilateral means and a significance level of 5%, a power of 95% was obtained.

Following the recommendations to calculate sample size for PCA analyses [18,19], a minimum of 5.8 to 6.2 cases were available for the analyses of each groups of variables (averages of the total week, weekday working hours, weekday non-working hours and weekend days). Each group included 13 to 14 variables related to the SB pattern and physical activity.

### 2.5. Statistical Methods

A descriptive study of the variables involved in the study and a bilateral comparison test of means between the group of patients with T2D and NGM were carried out to analyse whether there were any significant differences between them. Subsequently, the existence of correlations between the variables was studied in order to apply the multivariate technique of Principal Component Analysis (PCA).

PCA uses a vector space transformation to reduce the dimensionality of the data set into a smaller number of variables called principal components or dimensions (PCs) [20], which are lineal combinations of the original variables, where $C_{i}$ is the component i, $X_{j}$ the original variables and $\alpha_{i1}$ the weight of the variable $X_{j}$ in the component $C_{i}$.
(1)$$C_{i}=\alpha_{i1}\cdot X_{1}+\alpha_{i2}\cdot X_{2}+\cdots +\alpha_{ip}\cdot X_{p}=\overset{p}{\underset{j=1}{\sum }}\alpha_{ij}\cdot X_{j},1\leq i\leq p$$

The principal components are extracted in order of the contribution to the total variance of the data. The first principal components describe most of the variance of the data (more so when the original variables are more correlated). PCA is a useful technique to detect hidden patterns in the data. When the first few components contain the most important piece of information, the other components can be ignored. 

The PCA multivariate technique was applied to five groups, taking into account that SB patterns can vary depending on whether they are in weekends or weekdays and whether or not there are work commitments (working hours vs. non-working hours) [21]: (1). Variables in weekdays and weekends together; (2). Variables in weekdays; (3). Variables in weekends; (4). Variables in workdays and working hours; and (5). Variables in weekdays and non-working hours.

## 3. Results

### 3.1. Description of the Participants in Patterns of Physical Activity and SB: Differences between T2D and NGM

The men in the sample were on average 6.42 years older than the women, and their mean BMI was 3.76 units higher. Adults with T2D represented 38% of the total sample, 71.6% of whom were men. But if adults with diabetes older than 55 were selected, the percentage of men increased to 87.5%.

Table 1 presents descriptive statistics of variables related to occupational and habitual SB and physical activity, anthropometric variables and age. Significant differences were observed between the age and BMI variables in the adults diagnosed with T2D and the adults with NGM (Table 1). The adults diagnosed with T2D were on average ten years older than the NGM adults, and their BMI was 6.5 units higher. With regards to the physical activity pattern, there were no significant differences in the minutes/day of LIPA and MVPA between the T2D and NGM adults on weekdays. In contrast, there were significant differences (*p* = 0.0106) in the minutes/day of LIPA during the weekends, where the adults with T2D presented a significantly lower LIPA value (on average 13 min per day less) with respect to the adults with NGM. 

Regarding the differences in the SB pattern between the two groups, no significant differences were observed with respect to the minutes/day of total sedentary time, whereas there were significant differences in standing time (*p* = 0.0371) and total sedentary breaks/day (*p* < 0.001), both during weekdays and weekends. The patients with T2D on weekdays spent on average 29 min less standing time and took 32.7 fewer breaks than the adults with NGM. At weekends, the differences were 61.8 min and 21 breaks, respectively.

In addition, adults with T2D spent less time on average in short sedentary periods (less than 20 min) both during weekdays and weekends. With regards to working and non-working hours on weekdays, on average the number of breaks/days taken for periods less than 20 min during working hours in patients with T2D was 13 breaks/day less than the adults with NGM and 15 breaks/day less during non-working hours. These differences were significant (*p* < 0.001). During working hours, patients with T2D accumulated on average 39.3 min less in intervals of less than 20 min, while during non-working hours the difference was 30.9 min (Table 1). 

Although the number of breaks of more than 60-min periods was a small value (Table 1), the time spent in sedentary bouts >60 min was greater than the time spent in sedentary bouts between 20–40 and 40–60 min, for both patients with T2D and NGM. Both groups interrupted sedentary time more on weekdays than on weekends and during working hours than non-working hours. 

In relation to sedentary time, on average the adults with T2D remained seated for 11.8 min (total SB time/total SB breaks), compared to 7.6 min in adults with NGM. Also, the adults with T2D remained seated on average 4.4 min longer in periods of over 60 min. 

### 3.2. Correlation between Variables

#### 3.2.1. Correlation between Age, BMI and SB Pattern Variables

Age was positively related to BMI (Figure 1A): as age increased, the BMI value increased. Also, the SB accumulated in periods shorter than 20 min was negatively related to age and BMI. As age and BMI increased, less SB was accumulated in short periods of time. Contrarily, the SB accumulated in periods longer than 60 min was positively correlated with age and BMI (Figure 1A). The greater the age and BMI, the longer SB was accumulated in prolonged periods. On weekends, there was still a significant negative correlation between age, BMI and SB accumulated in periods of less than 20 min. As BMI and age increased, the accumulation of SB over prolonged periods of time increased.

#### 3.2.2. Correlation between the Variables That Measured the SB Pattern on Weekdays and at Weekends

When the correlation between the variables that measured the SB pattern on weekdays and at weekends was measured (Figure 1A), the variables were grouped by units of time, leading to the conclusion that the SB pattern did not vary between weekdays and weekends. The variables associated with units of time less than 20 min were grouped with the variables that measured total sedentary behaviour time, which confirmed the weight that the variables associated with units of less than 20 min had in the SB of both groups of patients. 

There was a negative correlation between the variables associated with time units less than 20 min and time units greater than 20 min; this negative correlation was greater with the variables associated with times longer than 60 min (Figure 1A).

#### 3.2.3. Correlation between the Variables That Measured the SB Pattern in Weekday Working Time and Weekday Non-Working Time

When studying therelationship between the SB pattern in weekday working time and weekday non-working time (Figure 1B), we observed patterns similar to those observed in the Section 3.2.2. One of the four groups was formed by the variables that measured SB in times less than 20 min, and another with those that measured SB in times longer than 60 min. In this second group were the BMI and age variables (as age increased, weight increased, the breaks increased and the bout time was longer) (Figure 1B). The other two groupswere made up of variables that measured SB in periods of 20–40 and 40–60 min, variables with little apparent relevance for the characterisation ofpeople with diabetes. Therefore, no relationship between SB during working and non-working hours was observed.

#### 3.2.4. Correlation between the Number of Breaks and the Time Spent in Sedentary Bouts

There was a positive and significant correlation between the variables that measured the breaks and the time spent in sedentary bouts in all time intervals (Figure 1C–F) in all contexts (weekdays, weekday working time, weekday non-working time and weekends).The slope of the least square lines (Appendix A) for periods less than 20 min was always greater for patients with T2D than adults with NGM, so the time between breaks, in this unit of time, was greater for patients with T2D. No differences were observed for time units greater than 20 min.

### 3.3. Principal Component Analysis

The study was carried out for five cases: (a) Variables that measured SB in weekdays and weekends together; (b) SB variables in weekdays; (c) SB variables in weekday working time; (d) variables in weekday non-working time; and (e) variables in weekends.

In the first case (a), the first two components explained 61.6% of the variability presented by the original data (Table 2). The variables with most weight in the first component were those that measured SB in periods under 20 min, followed by those that measured SB in periods greater than 60 min. In the grouping of patients (Table 2), a diffuse separation of patients with T2D and adults with NGM were observed with respect to the first two components.

In the case of using only the variables that measured SB during working hours and weekends (Table 2), the results were similar to those commented for the previous case.

Finally, SB during weekdays was stratified, based on the sedentary variables during working and non-working time. In the case of the variables that measured SB during non-working hours on working days, the T2D patients formed a cluster partially separated from NGM adults (Table 2). In this case, the variables that measured the SB pattern related to periods greater than 20 min contributed positively to the first component, while those related to periods less than 20 min had a negative impact (Figure 2A,C,D). The T2D patients were placed mainly on the right side of the scatter plot (Table 2), indicating a higher degree of SB. They were characterised by spending more time sitting and fewer interruptions in long time intervals outside working hours and higher BMI and age. The first two components explained 57.1% of the variability presented by the original data (Figure 2B).

## 4. Discussion

This study contributes to determining the differential SB pattern characteristic of patients with T2D and allows us to focus interventions to tackle SB in clinical practice. The SB pattern of patients with T2D was compared with adults with NGM using mean comparison tests and PCA-based multivariate analysis. The result makes it possible to establish a differentiated and characteristic SB pattern based on whether or not the patients have T2D and to suggest recommendations for interruption of the SB pattern in patients of working age with T2D. 

The main results of the study show that the adults with T2D have a lower number of total breaks in all contexts. In this regard, the SB pattern of the adults with T2D differs from the SB pattern of adults with NGM and is characterised by a lower number of breaks and accumulated time in short periods (less than 20 min) both during the week in working and non-working hours and during weekends. In longer periods of time there are no significant differences in general. The adults with T2D spend longer periods of sedentary time and break sedentary time less often than the adults without T2D, as other studies show [4,7]. However, we did not find any differences in MVPA between the two groups. Our findings provide information on the biological plausibility breaking sitting time might have for improving cardio metabolic risk profile in adults with T2D [22,23]. Results support the recommendation that patients with T2D, apart from regularly performing MVPA [24], should reduce sedentary time and interrupt sedentary time as much as possible, because these actions provide metabolic benefits independent of each other and of time spent in MVPA [6]. 

This study allows us to understand the SB patterns of patients with T2D in different contexts (total, occupational and free time) and the impact they have on total sitting time. Most studies have focused on a single context of sitting time (e.g., occupational) without being able to objectify the influence they might have in other contexts [25]. In the total sample of adults with office jobs, sedentary behaviour increases during free time on weekdays and at the weekend. The time spent sitting without interruptions during free time are significantly higher than on weekdays and during working hours. Other studies support our results reporting that adults with sedentary jobs are more likely to spend a lot of time sedentary during free time, [26] while others indicate that there is no relation between the amount of occupational sedentary time and the amount of time spent sitting during free time [27]. Our data show that there is no relation between sedentary time at work and during free time. Therefore, we cannot say that adults try to compensate for occupational SB with sedentary time during free time, nor that adults with more SB at work are also sedentary in their free time. According to our results, intervention strategies aimed at interrupting sitting time are necessary not only at work but in every context, and especially in free time, non-work time and weekends, as other studies conclude [28]. 

Through a multivariate PCA analysis, the people with T2D were characterised according to the pattern of accumulation of SB. The periods that make it possible to partially distinguish between adults with T2D and adults with NGM are short periods (less than 20 min). On the one hand, increasing the number of breaks in periods <20 min in adults with T2D is suggested. This result supports current evidence on the specific quantitative recommendations with regards to the frequency with which sitting time should be interrupted in the management of T2D. Current guidelines of the American Diabetes Association (ADA) state that sedentary periods for T2D patients should be interrupted at least every 30 min [5]. This recommendation is based on limited evidence (level C, the lowest grade) and it may be premature to suggest specific frequency recommendations [29]. 

On the other hand, given that the number of breaks/day in periods less than 20 min represent the majority of all the breaks accumulated during the day, we propose reaching a daily average of 70 breaks/day, 35 breaks/day during working hours and 35 breaks/day on weekdays during free time. From a practical perspective, the recommendation to reduce and interrupt sedentary time in different contexts, recommending a number of daily breaks depending on the context, may be more appropriate for these patients given that the exercise prescription is usually too short and intense to ensure good therapeutic adherence [8] and focuses the intervention only in a single context (leisure time). Also, since many people have office jobs and spend two-thirds of the day sitting down, with jobs that require longer periods of sitting down to achieve goals and not interfere with productivity, it might be beneficial to adapt the number of breaks to each individual based on physical characteristics and the social and work environment. Therefore, this study suggests that the recommendations should not only quantify the period of time that sedentary time should be interrupted, as most clinical trials focus on [29], but also the number of breaks accumulated/day in the different contexts to address SB in patients with T2D in a more comprehensive, effective and safe way. 

At present, SB detection and management tools in clinical practice are scarce (i.e., self-administered questionnaires) and have little applicability since they are complex to administer, they are unlikely to be integrated at the care level, they present medication errors and response biases, and they are not specific for patients with diabetes, which is why they do not identify characteristics of the SB pattern specific to T2D [30]. Furthermore, these questionnaires also do not detect the daily accumulation of sedentary time [31]. Quantifying the number of breaks/day and giving recommendations according to the different contexts would allow us to improve current standard care in the reduction of SB in people with T2D in a more targeted, effective and feasible way, providing a greater impact at the population level. 

The main strength of this study includes the objective measures of sedentary behaviour and physical activity in all contexts, including the workplace and leisure time. The instrument (the activPAL inclinometer) used to measure the SB pattern has shown good reliability and validity and is considered the gold standard for measuring SB, especially for measuring the accumulation of sedentary time throughout the day [16]. Another strength is the representativeness of the study sample; the prevalence of T2D is higher in men than women and increases with age [32]. In addition, adults with T2D who live in the community and have an office job were included. Most adults in Western nations spend most of their days employed [33]. It is especially important to focus the studies/interventions on adults with office jobs (jobs with little physical demand) given the magnitude and differentiated risk of a sedentary lifestyle in in this type of administrative jobs since they are the ones that report the highest levels (or highest prevalence) of sedentary lifestyle [25].

The main limitation of this study is that the analysis were cross-sectional, therefore causal relationships could not be examined. As a result, the need for new studies (clinical trials) aimed at evaluating the impact of the total number of breaks in people with T2D on the glycemic control variables should be noted. Our results may be useful to address SB in a targeted way for T2D patients with sedentary work and to develop new tools for detecting, evaluating and addressing SB in clinical practice. Another limitation is that the sample size of both groups is not equal as the recruitment for patients with diabetes had to be stopped due to the COVID-19 pandemic. These findings further support current guidelines promoting SB reduction in the prevention and management of type 2 diabetes [6,24]. In addition, all types of physical activity, including LIPA, should be promoted during free time to generate a positive balance with sitting time. Future tools and interventions aimed at adults with T2D doing office work could help to address SB, promote breaks from sedentary time and accumulate a minimum number of breaks/day to be achieved in short periods of sedentary time (<20 min), adapted to the different contexts.

In conclusion, adults with T2D had a sedentary behaviour pattern characterised by fewer number of breaks in time intervals of less than 20 min. On weekdays, a patient with T2D had 43.2 breaks on average, while adults with NGM had 74.3 breaks. In the case of breaks in time intervals greater than 20 min, significant differences were only observed during weekdays, where the average number of breaks was 8.3 for patients with T2D vs. 7.7 for adults with NGM. Effective interventions from clinical practice to tackle prolonged sedentary behaviour in office employees with T2D should focus on increasing the number of daily sedentary breaks.

Future research should also consider the use of experimental designs to evaluate the impact of the number of accumulated breaks on analytical parameters of glycemic control, mental health parameters and work productivity in patients with T2D, as well the design of new tools to detect and approach the SB pattern in these patients in clinical practice. 

## Figures and Tables

**Figure 1 ijerph-19-12245-f001:**
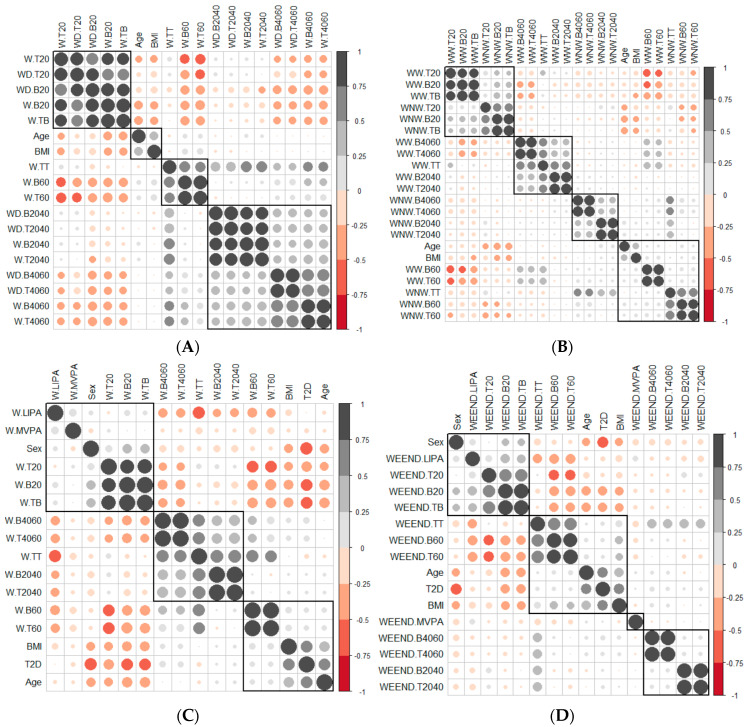
Correlation between the variables of the SB according to (**A**) Weekdays and at weekends, (**B**) Weekday working time and weekday non−working time, (**C**) Weekday total time, (**D**) Weekend, (**E**) Weekday working time and (**F**) Weekday non−working time. The gray and black circles indicate a positive correlation between variables, while the red circles indicate a negative correlation. When circles sizes is larger, then correlation between variables is greater. The groups of variables correlated with each other are marked with squares. T2D, diabetes type 2; BMI, body mass index; W.B20, weekday sedentary breaks <20 min (number/day); W−B2040, weekday sedentary breaks of 20−40 min (number/day); W−B4060, weekday sedentary breaks of 40−60 min (number/day); W.B60, weekday sedentary breaks >60 min (number/day); W.T20, weekday time spent in sedentary bouts <20 min (minutes/day); W.T2040, weekday time spent in sedentary bouts 20−40 min (minutes/day); W.T4060, weekday time spent in sedentary bouts 40−60 min (minutes/day); W.T60, weekday time spent in sedentary bouts >60 min (minutes/day); W.TB, weekday total sedentary breaks (number/day); W.TT, weekday total sedentary time (minutes/day) weekday; WD.B20, weekend sedentary breaks <20 min (number/day); WD.B2040, weekend sedentary breaks of 20−40 min (number/day); WD.B4060, weekend sedentary breaks of 40−60 min (number/day); WD.T20, weekend time spent in sedentary bouts <20 min (minutes/day); WD.T2040, weekend time spent in sedentary bouts of 20−40 min (minutes/day); WD.T4060, weekend time spent in sedentary bouts of 40−60 min (minutes/day); WW.B20, sedentary breaks <20 min during working hours (number/day); WW.B2040, sedentary breaks of 20−40 min during working hours (number/day); WW.B4060, sedentary breaks of 40−60 min during working hours (number/day); WW.B60, sedentary breaks >60 min during working hours (number/day); WW.T20, time spent in sedentary bouts <20 min during working hours (minutes/day); WW.T2040, time spent in sedentary bouts of 20−40 min during working hours (minutes/day); WW.T4060, time spent in sedentary bouts of 40−60 min during working hours (minutes/day); WW.T60, time spent in sedentary bouts >60 min during working hours (minutes/day); WW.TB, total sedentary breaks (number/day); WW.TT, total sedentary time (minutes/day) during working hours; WNW.LIPA, light−intensity physical activity during non-working hours (minutes/day); WNW.B20, sedentary breaks <20 min during non−working hours (number/day); WNW.B2040, sedentary breaks of 20−40 min during non−working hours (number/day); WNW.B4060, sedentary breaks of 40−60 min during non−working hours (number/day); WNW.B60, sedentary breaks >60 min during non−working hours (number/day); WNW.T20, time spent in sedentary bouts <20 min during non−working hours (minutes/day); WNW.T2040, time spent in sedentary bouts of 20−40 min during non−working hours (minutes/day); WNW.T4060, time spent in sedentary bouts of 40−60 min during non−working hours (minutes/day); WNW.T60, time spent in sedentary bouts >60 min during non−working hours (minutes/day); WNW.TB, total sedentary breaks during non−working hours (minutes/day); WNW.TT, total sedentary time (minutes/day) during non−working hours; WEEND.LIPA, weekend light−intensity physical activity (minutes/day); WEEND.B20, weekend sedentary breaks <20 min (number/day); WEEND.B2040, weekend sedentary breaks of 20−40 min (number/day); WEEND.B4060, weekend sedentary breaks of 40−60 min (number/day); WEEND.T20, weekend time spent in sedentary bouts <20 min (minutes/day); WEEND.T2040, weekend time spent in sedentary bouts of 20−40 min (minutes/day); WEEND.T4060, weekend time spent in sedentary bouts of 40−60 min (minutes/day); WEEND.TB, weekend total sedentary breaks (number/day).

**Figure 2 ijerph-19-12245-f002:**
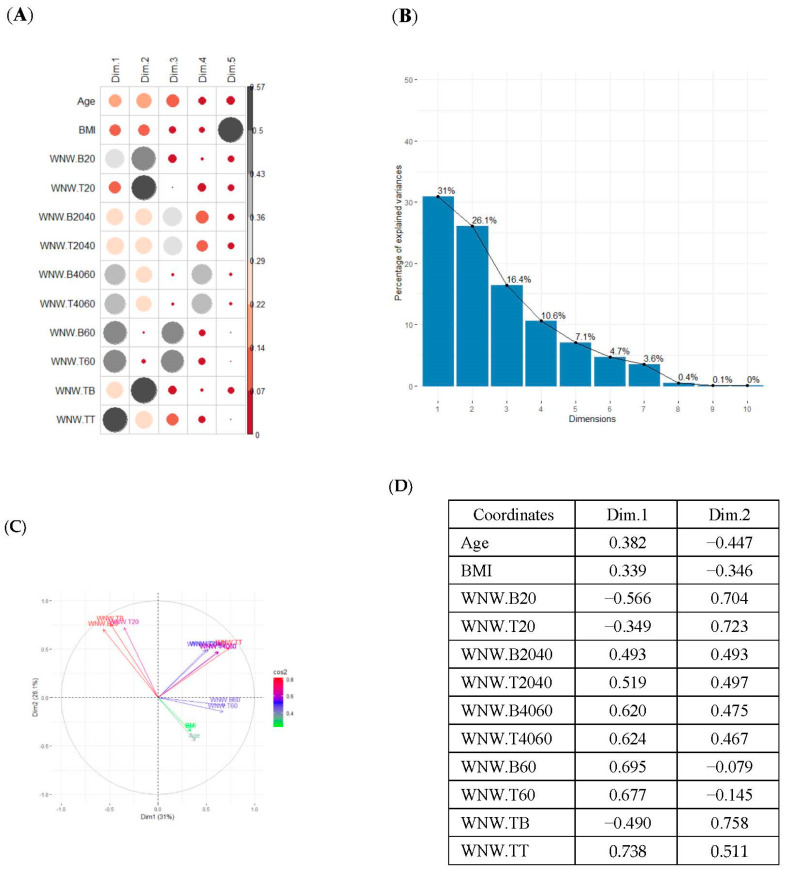
PCA results for the case of variables related to the SB pattern on weekdays and non−working hours: (**A**) correlation between the new components and original variables. The dimensions are the new components or variables results for PCA; (**B**) percentage of the original variability, explained by each of the new components; (**C**) Grouping of original variables, positive or negative contribution of the variables to the formation of the components; (**D**) Coordinates of the two new components for the case of variables related to weekday SB in non−working hours, the contribution of the original variables to the resulting components is described. BMI, body mass index; WNW.B20, sedentary breaks <20 min during non-working hours (number/day); WNW.B2040, sedentary breaks of 20−40 min during non−working hours (number/day); WNW.B4060, sedentary breaks of 40−60 min during non−working hours (number/day); WNW.B60, sedentary breaks >60 min during non-working hours (number/day); WNW.T20, time spent in sedentary bouts <20 min during non−working hours (minutes/day); WNW.T2040, time spent in sedentary bouts of 20−40 min during non-working hours (minutes/day); WNW.T4060, time spent in sedentary bouts of 40−60 min during non-working hours (minutes/day); WNW.T60, time spent in sedentary bouts >60 min during non−working hours (minutes/day); WNW.TB, total sedentary breaks during non−working hours (minutes/day); WNW.TT, total sedentary time (minutes/day) during non−working hours.

**Table 1 ijerph-19-12245-t001:** Description of the participants habitual and occupational SB, time spent in sedentary bouts, anthropometric and age variables. Mean, deviation and confidence interval of the difference between NGM and T2D participants.

	TYPE 2 DIABETES			NORMAL GLUCOSE METABOLISM			Between Groups T2D vs. NGM		
	Mean	SD	*n*	Mean	SD	*n*	95% CI		*p*-Value
Age (years)	55.8	6.3	77	45.2	8.9	128	−12.68	−8.52*	<0.001
BMI (kg/m^2^)	31.4	5.5	77	24.9	4.9	132	−7.98	−5.02*	<0.001
WEEKDAY—TOTAL TIME									
Light-intensity physical activity (minutes/day)	75.4	34.1	80	75.2	21.1	132	−8.45	8.05	0.9624
Moderate to vigorous physical activity (minutes/day)	43.7	27.2	79	47.7	21.3	132	−2.98	10.98	0.2656
Sedentary time (minutes/day)	572	144.8	75	599.3	101.8	132	−9.60	64.20	0.1518
Standing time (minutes/day)	273.4	106	80	302.4	80.6	132	2.14	55.86 *	0.0371
Total sedentary breaks (number/day)	50.3	17.2	75	82	28.4	132	25.49	37.85 *	<0.001
Sedentary breaks <20 min (number/day)	43.2	17	75	74.3	29.5	132	24.79	37.41 *	<0.001
Sedentary breaks of 20–40 min (number/day)	5	2.1	75	4.8	1.8	132	−0.74	0.38	0.5290
Sedentary breaks of 40–60 min (number/day)	1.7	1	75	1.5	0.8	132	−0.55	−0.33 *	0.0313
Sedentary breaks >60 min (number/day)	1.6	1.2	75	1.4	0.9	132	−0.56	0.05	0.1034
Time spent in sedentary bouts <20 min (minutes/day)	198.7	70.1	75	270.3	67.8	132	52.06	91.14 *	<0.001
Time spent in sedentary bouts 20–40 min (minutes/day)	138.9	56.3	75	133.6	49.4	132	−20.53	9.87	0.4954
Time spent in sedentary bouts 40–60 min (minutes/day)	84	47.3	75	71.1	38.6	132	−25.34	−0.34 *	0.0472
Time spent in sedentary bouts >60 min (minutes/day)	150.5	111.3	75	124.3	85.7	132	−55.14	2.80	0.0806
WEEKDAY—WORKING TIME									
Light-intensity physical activity (minutes/day)	37.5	26.2	53	24.5	11.9	115	−20.34	−5.66 *	0.0010
Moderate to vigorous physical activity (minutes/day)	20.7	14.6	25	18.4	11.4	98	−8.42	3.82	0.4692
Sedentary time (minutes/day)	283.8	129.6	47	294.4	64.5	115	−28.07	49.29	0.5950
Standing time (minutes/day)	2.3	1.2	53	1.9	0.9	115	−0.76	−0.04 *	0.0336
Total sedentary breaks (number/day)	28.5	15.2	46	41.7	19.1	115	7.70	18.84 *	<0.001
Sedentary breaks <20 min (number/day)	24.8	14.7	46	37.9	20	115	7.53	18.71 *	<0.001
Sedentary breaks of 20–40 min (number/day)	2.7	1.6	46	2.8	1.3	115	−0.4	0.63	0.6609
Sedentary breaks of 40–60 min (number/day)	0.8	0.7	46	0.7	0.6	115	−0.39	0.06	0.1633
Sedentary breaks >60 min (number/day)	0.6	0.7	46	0.4	0.5	115	−0.47	0.00	0.0513
Time spent in sedentary bouts <20 min (minutes/day)	116.4	65.7	46	153.7	53.6	115	16.10	58.61 *	0.0010
Time spent in sedentary bouts 20–40 min (minutes/day)	73.5	44.5	46	75.6	35.6	115	−12.23	16.45	0.7753
Time spent in sedentary bouts 40–60 min (minutes/day)	39.4	32.7	46	32.3	29.5	115	−17.99	3.64	0.1999
Time spent in sedentary bouts >60 min (minutes/day)	60.6	81	46	32.9	45.3	115	−52.47	−3.05 *	0.0327
WEEKDAY—NON-WORKING TIME									
Light-intensity physical activity (minutes/day)	45.3	25.9	68	49.6	15.9	115	−2.47	11.07	0.2187
Moderate to vigorous physical activity (minutes/day)	29	19.9	68	30.4	17.2	115	−4.29	7.01	0.6395
Sedentary time (minutes/day)	272.5	89.6	46	311.8	81.7	115	9.58	69.05 *	0.0119
Standing time (minutes/day)	165.9	83	68	184.5	49.5	115	−2.99	40.19	0.0963
Total sedentary breaks (number/day)	23.6	8.6	46	38.6	15.1	115	11.31	18.70 *	<0.001
Sedentary breaks <20 min (number/day)	19.6	8.5	46	34.6	15.4	115	11.24	18.66 *	<0.001
Sedentary breaks of 20–40 min (number/day)	2.2	0.8	46	2.2	1.0	115	−0.35	0.25	0.7487
Sedentary breaks of 40–60 min (number/day)	0.9	0.5	46	0.9	0.5	115	−0.18	0.17	0.9512
Sedentary breaks >60 min (number/day)	0.9	0.6	46	1.0	0.7	115	−0.11	0.33	0.3441
Time spent in sedentary bouts <20 min (minutes/day)	85.4	35.5	46	116.3	36.8	115	18.72	43.15 *	<0.001
Time spent in sedentary bouts 20–40 min (minutes/day)	62.1	22.5	46	61.7	29.0	115	−8.70	7.98	0.9334
Time spent in sedentary bouts 40–60 min (minutes/day)	41.3	24.3	46	41.7	26.2	115	−8.00	8.92	0.9154
Time spent in sedentary bouts >60 min (minutes/day)	83.8	66.1	46	92.1	69.9	115	−14.60	31.15	0.4825
**WEEKEND**									
Light-intensity physical activity (minutes/day)	67.3	36.2	77	80	30.4	132	3.14	22.26 *	0.0106
Moderate to vigorous physical activity (minutes/day)	34.9	31.1	73	33.9	33.1	132	−10.05	8.05	0.8297
Sedentary time (minutes/day)	548.3	135.1	73	518.2	117	132	−66.77	6.58	0.1120
Standing time (minutes/day)	235.3	96.8	73	297.1	97.3	132	34.22	89.38 *	<0.001
Total sedentary breaks (number/day)	38.3	12	73	59.3	27.2	132	15.62	26.36 *	<0.001
Sedentary breaks <20 min (number/day)	31.9	12.3	73	52.5	27.8	132	15.14	26.12 *	<0.001
Sedentary breaks of 20–40 min (number/day)	3.8	1.7	73	3.6	1.7	132	−0.74	0.21	0.2848
Sedentary breaks of 40–60 min (number/day)	1.6	1	73	1.4	1.0	132	−0.54	0.02	0.0729
Sedentary breaks >60 min (number/day)	2.1	1.3	73	1.9	1.0	132	−0.59	0.11	0.1840
Time spent in sedentary bouts <20 min (minutes/day)	151.2	61.6	73	166.8	64.7	132	−2.28	33.39	0.0910
Time spent in sedentary bouts 20–40 min (minutes/day)	109.4	46.9	73	99.7	46.5	132	−37.03	17.60	0.4981
Time spent in sedentary bouts 40–60 min (minutes/day)	79.1	49.5	73	67.1	46.9	132	−25.89	1.75	0.0908
Time spent in sedentary bouts >60 min (minutes/day)	208.5	134.4	73	184.6	113.9	132	−60.13	12.38	0.2015

T2D, diabetes type 2; NGM, normal glucose metabolism; SD, standard deviation * There are statistically significant differences.

**Table 2 ijerph-19-12245-t002:** Scatter plots of patients with T2D and adults with NGM based on the components (dimensions) obtained with PCA for the different groups of variables that measure SB.

PCA Variables	Variability Explained by the First Two Components	Grouping of Patients Based on the 2 Dimensions Obtained with PCA	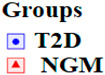
Weekday and weekend	61.6%	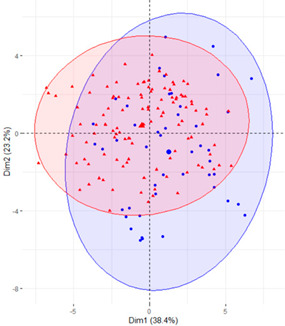	
Weekday	61%	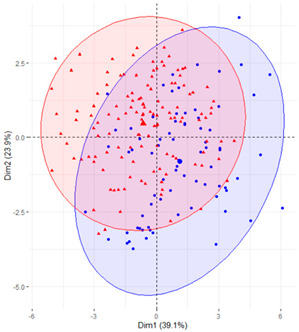	
Weekday working time	63.1%	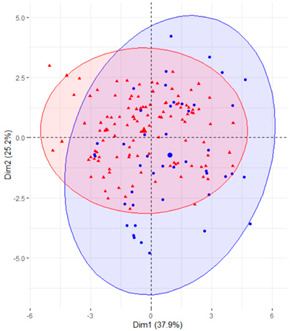	
Weekday non-working time	57.1	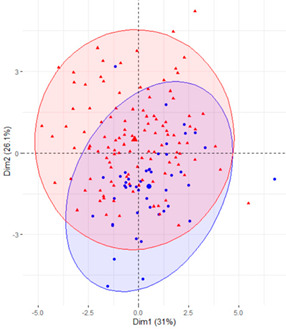	
Weekend	54.7	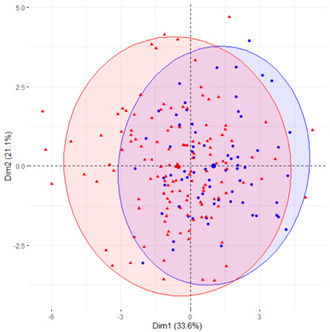	

T2D, type 2 diabetes; NGM, normal glucose metabolism; PCA, principal component analysis.

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
