# Peer review of "Differences in Free-Living Patterns of Sedentary Behaviour between Office Employees with Diabetes and Office Employees without Diabetes: A Principal Component Analysis for Clinical Practice"

_ijerph, 2022, doi:10.3390/ijerph191912245_

Round 1

Reviewer 1 Report

This is a well-written and well-illustrated manuscript determining the differential sedentary behaviour (SB) pattern characteristic of patients with T2Diabetes (T2D). For this, the authors used Principal Component Analysis (PCA) to examine ActivPal-based free-living patterns of SB in office employees with T2D compared with office employees without diabetes. Overall, the report is well organized. Yet, some changes should be reconsidered.

Minor revisions:

·         Abstract:

·         The details put in the conclusion should rather be in the results part instead

·         The conclusion should be reformulated

·         These two parts (results and conclusion) should be reorganized

·         Introduction:

·         The authors should briefly introduce/define PCA and ActivPal before announcing the objectives

·         Methods:

o   The parts study design and participants should be merged

·         Results: the authors should use the past tense

·         Conclusion: the paragraph “Finally, future research should consider the use of experimental designs to evaluate the impact of the number of accumulated breaks on analytical parameters of glycemic control, mental health parameters and work productivity in patients with T2D, as well the design of new tools to detect and approach the SB pattern in these patients in order to address T2D in clinical practice more effectively.” should be put at the end of the manuscript (as a perspective).

Other Minor changes suggested:

·         Page2 line 69-71 and 79-80: correct the font.

·         Page2 line 79-87: “Given the need to provide a comprehensive understanding of free-living patterns of SB in clinical populations, the aim of this study was to use PCA to examine ActivPal-based free-living patterns of SB in office employees with T2D compared with office employees without diabetes in order to (i) facilitate the identification of differences and similitudes between adults with and without diabetes to best characterise the free-living patterns of SB in adults with T2D; (ii) help primary care providers focus on SB patterns tailored to this clinical population; and (iii) facilitate the identification of potential targets for behavioural interventions on SB for T2D, including frequency of interruptions and length of sedentary bouts.” è too long sentence, should be split.

·         Page 6, line 187-188: “But if adults with diabetes older than 55 are selected, the percentage of men increased to 87.5%”. è “But if adults with diabetes older than 55 were selected, the percentage of men increased to 87.5%”.

·         Page 6, line 189-190: correct “Table 2 present descriptive statistics of variables related to occupational and habitual SB and physical activity, anthropometric variables and age.”è “Table 2 presents/recapitulates descriptive statistics of variables related to occupational and habitual SB and physical activity, anthropometric variables and age.”

·         Page9, line 229-236: use past tense for the results

·         Page 11, line 273-374: correct the font

·         Page 14, line 330-331: correct the font

·         Page 14, line 321-333: results should be in past tense.

·         Page 15, line 349: correct the font

Author Response

Barcelona, 11th of September 2022

Dear reviewer

It is with excitement that we submit a revised version of the manuscript ijerph-1860295 “Differences in Free-Living Patterns of Sedentary Behaviour Between Office Employees with and without Diabetes Type 2: A Principal Component Analysis for Clinical Practice” 

We appreciate the comprehensive comments and feedback from the reviewers. The revisions, based on the co-authors’ collective input, have carefully considered all comments and includes several changes which we agree benefits the manuscript. We have responded specifically to each suggestion below.

Best wishes,

Francesc Alòs

Reviewer 2 Report

see attachment

Author Response

(The authors gave the same response as above.)

Reviewer 3 Report

The research paperDifferences In Free-Living Patterns Of Sedentary Behaviour 2 Between Office Employees With And Without Diabetes Type 2: A Principal Component Analysis For Clinical Practice has certain merits for publication in this journal. However, there are certain points that should be worked before it is considered for publication:  

1. Title should be corrected as its not a norm to use preposition and conjunction starting from a capital letter in the title. Check “In” “of” and “And”

2. Add conclusion statement at the end of abstract

3. Check the line-spacing from paragraph starting from “Characterising the SB pattern in patients with …...in introduction section.  

4. How the authors will explain the casual or habitual sedentary behavior and T2DM related SB?

5. Authors should consider adding abbreviation in tables’ footnotes.

6. Paper has font size problems at various stages. For example legend of figure 1.

7. Legend of figure 1 and figure 2 should explain what the figure is suggesting and what software (if any) is used to generate this figure along with statistical analysis.

Author Response

(The authors gave the same response as above.)

Reviewer 4 Report

English language needs more corrections. The text and information are very ambiguous.  Principal Component Analysis is a statistical analysis. It should not include in the title. I studied the manuscript several times, but I did not reach to any novelty or solve a new problem. Increase in activity in time of working is effective on glucose control. This is a fact. In other word, significance of content is under question. Ethical approval is not identified. The study was approved by the ethics committee of each hospital and is referenced. This is the ethical number; P18/102? I think it is wrong. 

please highlight the novelty and importance of the manuscript and re-write more understandable.

Author Response

(The authors gave the same response as above.)
